# Disposable Molecularly Imprinted Polymer-Modified Screen-Printed Electrodes for Rapid Electrochemical Detection of l-Kynurenine in Human Urine

**DOI:** 10.3390/polym16010003

**Published:** 2023-12-19

**Authors:** Roberta Del Sole, Tiziana Stomeo, Lucia Mergola

**Affiliations:** 1Department of Engineering for Innovation, University of Salento, Via per Monteroni Km 1, 73100 Lecce, Italy; lucia.mergola@unisalento.it; 2Center for Bio-Molecular Nanotechnology, Istituto Italiano di Tecnologia, Via Barsanti 14, 73010 Arnesano, Italy; tiziana.stomeo@iit.it

**Keywords:** l-kynurenine, electrochemical molecularly imprinted polymer, poly(*o*-phenylenediamine), modified screen-printed electrode, differential pulse voltammetry, human urine

## Abstract

l-Kynurenine (l-Kyn) is an endogenous metabolite produced in the catabolic route of l-Tryptophan (l-Trp), and it is a potential biomarker of several immunological disorders. Thus, the development of a fast and cheap technology for the specific detection of l-Kyn in biological fluids is of great relevance, especially considering its recent correlation with SARS-CoV-2 disease progression. Herein, a disposable screen-printed electrode based on a molecularly imprinted polymer (MIP) has been constructed: the *o*-Phenylenediamine monomer, in the presence of l-Kyn as a template with a molar ratio of monomer/template of 1/4, has been electropolymerized on the surface of a screen-printed carbon electrode (SPCE). The optimized kyn-MIP-SPCE has been characterized via cyclic voltammetry (CV), using [Fe(CN)_6_)]^3−/4−^ as a redox probe and a scanning electron microscopy (SEM) technique. After the optimization of various experimental parameters, such as the number of CV electropolymerization cycles, urine pretreatment, electrochemical measurement method and incubation period, l-Kyn has been detected in standard solutions via square wave voltammetry (SWV) with a linear range between 10 and 100 μM (R^2^ = 0.9924). The MIP-SPCE device allowed l-Kyn detection in human urine in a linear range of 10–1000 μM (R^2^ = 0.9902) with LOD and LOQ values of 1.5 and 5 µM, respectively. Finally, a high selectivity factor α (5.1) was calculated for l-Kyn toward l-Trp. Moreover, the Imprinting Factor obtained for l-Kyn was about seventeen times higher than the IF calculated for l-Trp. The developed disposable sensing system demonstrated its potential application in the biomedical field.

## 1. Introduction

In the last few years, the catabolic route of the essential amino acid l-Tryptophan (l-Trp) through the kynurenine pathway (KP) has attracted numerous research activities due to its central role in a huge variety of pathophysiological processes. Three different enzymes catalyze the overall KP: indoleamine 2,3-dioxygenase 1 (IDO1), indoleamine-2,3-dioxygenase 2, and tryptophan 2,3-dioxygenase. l-Kynurenine (l-Kyn) is stable and is the main catabolite of l-Trp in the human body, synthesized during the catabolic route promoted by the IDO1 enzyme through the initial degradation of l-Trp to formylkynurenine, which is rapidly transformed into l-Kyn. IDO1 is expressed in various cell types, leading to KP activation in different tissues and cells, like dendritic cells, macrophages, and tumor cells [1]. The high immunological and neurological activity of l-Kyn enables it to inhibit T-cell proliferation, to reduce the activity of natural killer cells, to spread the immunological tolerance among dendritic cells, and to promote the differentiation of regulatory T-cells. KP has also been involved in neurodegenerative and psychiatric disorders [2,3]. Moreover, l-Kyn is a ligand of the aryl hydrocarbon receptor involved in various physiological functions, such as tumor invasion and/or migration [1,4]. Tumoral cells, through the overexpression of the IDO1 enzyme, greatly enhance l-Trp catabolism via the KP for escaping immune surveillance [4]. Several IDO1 inhibitors are currently in clinical trials for the treatment of cancer [5]. Consequently, an increase in the l-Kyn amount with respect to physiological values in the human body is a signal of an infection status and the detection of the l-Kyn amount as well as the l-Kyn to l-Trp ratio is considered an important and innovative diagnostic tool to have information about the efficacy of novel chemotherapeutics targeting the KP and about the progression of the malignancy [6]. The crucial role of l-Kyn in biomedical research is also confirmed by the very recent studies on the correlation of l-Kyn levels in human fluids and SARS-CoV-2 disease progression. For instance, Bizjak et al. demonstrated that l-Kyn can be a suitable biomarker in the detection of inflammatory and hyperinflammatory conditions of the SARS-CoV-2 disease in the acute and long-term progression as well as a chronic subclinical systemic inflammation characteristic for long COVID and more evidence for the post-COVID condition [7].

Up to now, high-performance liquid chromatography coupled with various detectors, mostly UV or MS detection and sometimes fluorescence or electrochemical detection, has been the main traditional and frequently used method for l-Kyn detection in biological tissues and fluids [8,9,10,11]. Even if high selectivity and sensitivity have been obtained, various drawbacks, such as the time consumption, high instrumentation costs, great technician expertise needed, bulky equipment, and long sample pretreatment characteristic of the above techniques, has led many researchers to move towards alternative, innovative, and feasible detection methods. Other techniques, such as gas chromatography, immunoassays, and, more recently, sensors, have also been used. Papers that describe sensors technique for l-Kyn detection without using chromatographic processes have been published starting from 2017 and show various advantages, especially in terms of high sensitivity and the potentiality to reduce the size of the system to obtain a portable device [12,13,14,15,16,17]. Among them, electrochemical sensors have been demonstrating interesting successes and/or potentiality, such as an elevated sensitivity, rapid response, reduced size of apparatus, simple installation, easy sample preparation, and in situ analysis. Recently, Sadok and co-workers prepared for the first time an electrochemical sensor for l-Kyn detection using a nafion-coated carbon electrode and applying a differential pulse adsorptive stripping voltammetry. The developed method was successfully applied for l-Kyn evaluation in culture medium obtained from ovarian carcinoma cells [16]. In another work, a specific chemosensor (3-formyl-4-(chloro)-7-(diethylamino)-coumarin) was used to prepare a fluorescent-based assay for l-Kyn detection in synthetic urine samples [17].

One of the limitations of the electrochemical techniques is the lack of selectivity when other electroactive compounds are also present in the matrix. However, molecularly imprinted polymers (MIPs) have been successfully employed for the selective recognition of numerous small molecules, demonstrating that they are an effective route to solve the above need. MIPs are highly crosslinked polymers based on the creation of specific interactions between a template (analyte molecule) and a functional monomer, which is polymerized in the presence of the template. After polymerization, the template is removed, leaving in the polymeric matrix specific recognition sites complementary in shape, size, and chemical functions to the template, which can be specifically linked and detected successively during the recognition process [18,19,20,21]. MIPs are synthetic smart polymers that possess various advantages, such as a high physical robustness, strength, and resistance to elevated temperatures and pressures and inertness towards acids, bases, metal ions, and organic solvents, along with a high stability over time and low production costs.

In our previous work, an MIP specific for l-Kyn was developed and employed as a sorbent for an easy l-Kyn purification on a solid phase extraction system from biological fluids [22]. Various papers on MIP-based sensors have been published, mainly with electrochemical or optical detection [23].

The screen-printed electrode (SPE) is a well-established technology consisting of the fabrication of miniaturized electrochemical sensing devices that can be easily integrated with MIP technology to confer higher specificity and sensitivity to the SPE when the analysis is conducted in complex matrices. The SPE has various advantages compared to traditional analytical methods, which allowed it to gain high diffusion into the analytical field. In fact, the SPE is a low-cost disposable miniaturized electrode that can be easily integrated with lab-on-a-chip and microfluidic analysis systems [24,25]. Different MIP-based SPEs were recently prepared for the selective detection of organic compounds, such as drugs [26,27,28], tumoral biomarkers [29,30,31], and environmental pollutants [32,33,34]. An imprinted electrochemical sensor for ecstasy detection was obtained through electropolymerization on a screen-printed carbon electrode (SPCE) of *o*-phenylenediamine (*o*-PD), used as a functional monomer, and successfully applied for the selective detection of the template in human blood serum and urine [27]. Karami and co-workers prepared an MIP-based biosensor for prostate-specific antigen and myoglobin simultaneous detection in biological matrices, modifying a gold SPE. In particular, the working electrode was electropolymerized by using acrylamide as functional monomer and N,N′-methylenebisacrylamide as crosslinking agent [29]. A gold SPE was also modified via the electropolymerization of poly(m-phenylenediamine) to obtain a selective sensor for sulfamethizole detection in environmental matrices [33].

In the present work, for the first time, a combination of SPE electrochemical sensors and MIP advantages has been reported with the aim to obtain a disposable and portable device for specific l-Kyn recognition. Some key aspects have been optimized, such as the choice of a suitable and effective imprinted polymer and also a reliable new analytical method based on square wave voltammetry (SWV) for urine sample analysis of l-Kyn as well an easy urine pretreatment. The SPE was used as a sensor for specific l-Kyn recognition after its modification through the electrochemical in situ MIP synthesis. *o*-PD monomer electropolymerization on the working electrode surface of a commercial SPCE in the presence of an l-Kyn molecule as a template was carried out. The optimized modified sensor specific for l-Kyn detection (kyn-MIP-SPCE) has been characterized via cyclic voltammetry (CV), using [Fe(CN)_6_)]^3−/4−^ as a redox probe, and scanning electron microscopy (SEM). After the optimization of various experimental parameters, such as the number of CV electropolymerization cycles, electrochemical measurement method, and incubation period, l-Kyn was first detected in a standard solution via SWV. The optimized analytical procedure was then performed for l-Kyn detection in urine samples. A pretreatment urine optimization was also carried out.

## 2. Experimental

### 2.1. Materials

l-Kyn, l-Trp, acetic acid, and *o*-PD were supplied from Sigma-Aldrich (Steinheim, Germany, www.sigmaaldrich.com accessed on 18 November 2023). Analytical grade methanol (MeOH) was obtained from J. T. Baker (Deventer, Holland, www.jtbaker.com accessed on 18 November 2023). In addition, 0.1 M phosphate buffer solutions (PBSs, pH 5 and pH 7.4) were prepared using Na_2_HPO_4_ and NaH_2_PO_4_, both purchased from Sigma-Aldrich. All solutions were prepared using ultrapure water obtained with a water purification system (Human Corporation, Songpa-gu, Republic of Korea, www.humancorp.co.kr accessed on 18 November 2023). Commercial SPCE DRP-110 (Metrohm Co., Schiedam, The Netherlands, www.metrohm.com accessed on 18 November 2023) with a 4 mm carbon working electrode, carbon-based counter electrode, and silver pseudo-reference electrode was used. Potassium hexacyanoferrate (III) (K_3_[Fe(CN)_6_]), potassium hexacyanoferrate (II) trihydrate (K_4_[Fe(CN)_6_]^.^3H_2_O), and KCl for probe experiments were supplied from Merck (Darmstadt, Germany, https://www.merckgroup.com accessed on 18 November 2023). H-PTFE syringe filters (0.2 μm) were supplied from J.T.Baker^®^, and Oasis^®^ HLB VAC RC (60 mg) cartridges were purchased from Waters (Milford, MA, USA, https://www.waters.com accessed on 18 November 2023). Human urine samples were obtained from healthy volunteers.

### 2.2. Apparatus

CV and SWV were performed using a Metrohm Autolab PGSTAT204 potentiostat (Metrohm AG, Herisau, Switzerland). NOVA 2.1 software was used for data acquisition. All electrochemical measurements were conducted at room temperature (22 °C). Morphological analyses were conducted using a scanning electron microscope (SEM) on a Dual Beam FIB/SEM HeliosNanoLab600i instrument (Fei company, Hillsboro, OR, USA). The electron beam current was set to 0.34 nA with an accelerating voltage of 5 kV.

### 2.3. Preparation of kyn-MIP-SPCE

Bare SPCEs were initially washed with ultrapure water for activation and then dried at 40 °C for 2 h.

kyn-MIP-SPCE was prepared as follows. Firstly, a self-assembly mixture was obtained by mixing 1 mL of l-Kyn (4 mM) and 1 mL of o-PD (1 mM) in PBS 0.1 M (pH 5), for 20 min at 22 °C. Then, a 40 μL drop of prepolymerization solution was added onto the SPCE surface and the electropolymerization was carried out performing a CV from 0.0 to +1.2 V with a scan rate of 100 mV/s during 5, 15, 20, or 25 cycles. All modified sensors were washed firstly in water and after in a mixture of ultrapure water and MeOH (1/1, *v*/*v*) for 10 min to remove the template molecule l-Kyn. Finally, after a further washing in ultrapure water, all modified sensors were dried at 40 °C for 24 h before use. Non-imprinted film was electropolymerized on the SPCE using the same procedure described above but in the absence of l-Kyn in the prepolymerization mixture (NIP-SPCE).

### 2.4. Electrochemical Measurements of l-Kyn Standard Solutions Using kyn-MIP-SPCE

Electrochemical measurements were performed by incubating the modified sensor with l-Kyn solutions, prepared in PBS 0.1 M at pH 5 (as supporting electrolyte), at different concentrations ranging from 10 to 100 µM at 22 °C. Then, 50 μL of l-Kyn solutions was analyzed by drop-casting it onto the kyn-MIP-SPCE working electrode surface and subjected to SWV in the potential range from 0.0 to +1.6 V (frequency of 25 Hz) with a pulse amplitude of 20 mV and step potential of 5 mV. The peak current (I_P_) was measured at around 0.7 V for the oxidation of l-Kyn, after baseline correction by the ‘moving average’ function present in Nova 2.1 software. All experiments were repeated three times.

The modified electrode was characterized via CV, performed between −0.2 and 1.0 V (scan rate of 100 mV), and 50 μL of the redox probe [Fe(CN)_6_)]^3−/4−^ (4 mM) prepared in KCl 3 M was added onto the working electrode, to monitor all construction steps.

### 2.5. Selectivity Studies

For selectivity studies, kyn-MIP-SPCE and the corresponding NIP-SPCE were incubated with l-Trp and l-Kyn solutions (25 μM) prepared in PBS 0.1 M (pH 5). SWV analysis was performed in the potential range of 0.0 to +1.6 V in the same conditions previously mentioned, and the I_P_ values were registered and compared to evaluate the selectivity factor (α) and the Imprinting Factor (IF), calculated according to Equations (1) and (2):α = ΔI(Kyn)/ΔI(Trp)(1)
IF = ΔI (MIP)/ΔI(NIP)(2)
where ΔI(Kyn) and ΔI(Trp) represent the difference between the peak current of the supporting electrolyte and the I_P_ registered after incubation for 6 min of kyn-MIP-SPCE with l-Kyn and l-Trp solutions.

ΔI(MIP) and ΔI(NIP) represent the difference between the peak current of the supporting electrolyte and the I_P_ obtained after incubation for 6 min of each analyte with kyn-MIP-SPCE and NIP-SPCE. All experiments were repeated three times.

### 2.6. Human Urine Sample Preparation

Human urine samples were obtained from healthy volunteers (age 40–48 years). Fresh human urine was refrigerated for 3 h at 4 °C. Then, it was filtered with a PTFE-H syringe filter (0.2 μm), spiked with increasing amounts of l-Kyn to reach final concentrations ranging from 10 to 1000 µM, and finally diluted (1/10, *v*/*v*) with PBS 0.1 M pH 7.4.

Then, 50 μL of each dilution were added onto the modified SPCE working electrode for electrochemical measurements. All experiments were repeated three times.

### 2.7. Electrochemical Experiments in Human Urine

Electrochemical measurements were performed by incubating the modified sensor with different concentrations of spiked urine (10, 50, 100, 400, and 1000 µM). In particular, 50 μL of all spiked solutions were deposited by drop-casting onto the working electrode of the kyn-MIP-SPCE and subjected, after 10 min, to SWV in the potential range from 0.0 to +1.6 V (frequency of 25 Hz) with a pulse amplitude of 20 mV and step potential of 5 mV. All experiments were repeated three times.

### 2.8. Method Validation

The optimized method previously described was validated by evaluating the linearity range (LR), correlation coefficient (R2), Limit of Detection (LOD), and Limit of Quantification (LOQ) of l-Kyn in the human urine of healthy donors. Additional rebinding experiments were conducted and the LOD and LOQ were calculated from the linear regression analysis of the concentration measured from the modified sensor using the following equation:LOD = 3 × DSx/y/b(3)
LOQ = 10 × DSx/y/b(4)
where DSx/y represents the standard deviation of the regression and b represents the slope of the regression line.

## 3. Results and Discussion

### 3.1. Preparation and Characterization of kyn-MIP-SPCE

In a previous work, we prepared an imprinted polymer for l-Kyn recognition using methacrylic acid and trimethylpropane trimethacrylate as a functional monomer and crosslinker, respectively, and excellent recognition properties were found when the MIP was used as sorbent for the solid phase extraction of urine samples [22]. In the present work, a first attempt to use the same MIP to modify the working electrode of an SPCE was made. However, to overcome some drawbacks typical of MIP sensors obtained from the deposition of a conventional polymer, such as solvent consumption and the difficulty in controlling the thickness, morphology, and reproducibility to have an effective electron transfer interface, we addressed our research towards the direct MIP synthesis on the work electrode surface through the electropolymerization technique.

*o*-PD has been the most used functional monomer for MIP-based sensor fabrication through the electropolymerization process thanks to its high stability [27,30,33]. Moreover, the presence of free amine groups in its structure favors the formation of interactions with the template during the prepolymerization step and the creation of specific cavities for l-Kyn rebinding in the polymeric film layer.

After the *o*-PD-l-Kyn complex formation, a drop of the mixture was deposited onto the SPCE surface and subjected to electropolymerization via CV in a potential range between 0.0 and +1.6 V and with a scan rate of 100 mV/s (Figure 1). The choice of the scan cycle number is important during modified sensor construction since it permits the modulation of the thickness of the polymeric layer on the working electrode and, consequently, its performance.

To this aim, four different electrodes were prepared using different scan cycles (5, 15, 20, 25) to evaluate the right condition to improve the sensitivity of the sensor. In the first scan cycle (Figure 2), the presence of a typical oxidation peak of *o*-PD can be observed that occurs at +0.26 V with a peak current (I_P_) of 4.68 × 10^−5^ A, along with the absence of a reduction peak that confirms the irreversibility of *o*-PD electropolymerization and, therefore, the success of the polymerization process. As can be observed, when the number of scan cycles increased, the electrode surface was covered with a thick layer of polymeric film, reducing the conductivity of the sensor with a current near to zero (Figure 2b).

After the washing steps to remove the template, each modified sensor was tested by drop casting 50 μL of a 100 μM l-Kyn solution (0.1 M, PBS pH 5) onto the working electrode and performing a SWV analysis. The sensitivity of the sensors decreases as the number of scan cycles increases. In detail, the peak current intensity (I_p_) increases for five scans (8.5 μA) compared to bare SPCE (2.6 μA), confirming the presence of specific binding sites into the polymeric matrix, which are close to the sensor surface, and aiding charge transfer. Then, it gradually decreases when 15, 20, and 25 cycles are tested (from 3.7 μA to 1.5 μA) since a higher thickness of the film makes the template sites farther from the sensor surface, which probably causes a greater hindrance to the charge transfer. The best performance was obtained with the kyn-MIP-SPCE prepared with five scan cycles, which was chosen for further studies.

All sensor construction steps were characterized via the CV method using [Fe(CN)_6_)]^3−/4−^ as a redox probe for monitoring changes in the electron transfer. In detail, a drop (50 μL) of redox probe solution 4 mM prepared in KCl 3 M was placed on a bare SPCE and on kyn-MIP-SPCE before washing, after washing, and after rebinding with l-Kyn. As expected, the curve obtained from the bare SPCE (Figure 3a) presents two typical redox peaks of the ferricyanide probe. In the b curve (Figure 3b), it can be noted that the absence the of I_P_ is associated with the redox probe, due to the growth of a non-conductive polymeric layer that hinders the electron transfer. The elution of l-Kyn during the washing step leaves empty cavities in the polymer that facilitate the access of the probe on the electrode surface, causing an increase in the I_P_ compared to curve b even if the presence of the polymeric layer does not allow it to return to the high I_p_ found for bare SPCE (Figure 3c). Finally, after rebinding, the presence of l-Kyn bound to some cavities reduces the access of the probe on the working electrode, causing a slight decrease in the I_p_ compared to curve c [28] (Figure 3d).

The morphological characterization of modified SPCE was verified using a scanning electron microscope (SEM) on a Dual Beam FIB/SEM HeliosNanoLab600i instrument. Figure 4 shows the in-plane SEM images of bare SPCE (Figure 4a–d), kyn-MIP-SPCE (Figure 4b–e), and NIP-SPCE (Figure 4c–f) acquired at two different magnifications. The bare SPCE revealed the presence of corrugations, more visible at higher magnification (Figure 4d), typical of the carbonaceous coating in the commercial working electrode. The morphology of the modified screen-printed electrodes, kyn-MIP-SPCE and NIP-SPCE, is shown in Figure 4e and 4f, respectively. Compared with the bare SPCE (Figure 4d), a slight attenuation of roughness due to the formation of a thin polymeric layer on the working electrode after the electropolymerization process is evident. The obtained images show that the kyn-MIP-SPCE and NIP-SPCE have a round-shaped morphology. As expected, no significant differences were observed when comparing the imprinted and non-imprinted sensors.

### 3.2. Analytical Performance of kyn-MIP-SPCE

kyn-MIP-SPCE was tested with different concentrations of l-Kyn (10–100 µM) via SWV analysis, a very sensitive electroanalytical technique widely used in electrochemical analysis. To evaluate the time of incubation necessary to obtain a stable signal, kinetic studies were performed following the I_p_ during 10 min. In Figure 5, kinetic curves, obtained by incubating the modified sensor with different concentrations of l-Kyn standard solutions, were reported. As can be seen, after 2 min the current became stable, and a plateau was reached. The graph also reported the kinetic performance of a bare SPCE, incubated with 10 μM of l-Kyn standard solution that demonstrated the effect in terms of sensitivity of the presence of the imprinted polymeric layer electropolymerized on the sensor. Considering this result, all following analyses were performed incubating all standard solutions for 6 min to ensure the stability of the I_p_.

A first attempt to evaluate the performance of the modified sensor was made by loading l-Kyn standard solutions onto the working electrode at different concentrations. The inset in Figure 6 shows the SWV current response of the kyn-MIP-SPCE in the presence of different concentrations of l-Kyn standard solutions prepared in PBS 0.1 mM (pH 5). Analyzing the data after 6 min of incubation, a direct proportionality between I_p_ and l-Kyn concentrations was achieved with a correlation coefficient equal to 0.9924 (Figure 6).

### 3.3. Selectivity Studies

To evaluate the selectivity of the modified sensor prepared towards l-Kyn, kyn-MIP-SPCE was also incubated with a solution of l-Trp, which was chosen as an interferent, and the selectivity factor α was calculated. The results obtained demonstrated that l-Kyn generates an electrochemical response higher than l-Trp, and this is due to the presence in the polymeric film of specific cavities complementary in size, shape, and chemical composition to l-Kyn that were specifically retained through the formation of hydrogen bonds with hydrogen and nitrogen atoms of *o*-PD.

The low ΔI recorded after l-Trp incubation with kyn-MIP-SPCE was due to the formation of non-specific hydrogen bonds between l-Trp and the surface of the polymeric layer. Indeed, the selectivity factor α equal to 5.1 of kyn-MIP-SPCE calculated toward l-Trp was very high, demonstrating the high selectivity of the modified sensor for l-Kyn. Comparing this last result with another MIP for l-kyn reported in the literature [22], a similar selectivity factor was obtained (α = 4), demonstrating that each developed system is suitable for l-Kyn recognition in its specific application.

l-Kyn and l-Trp solutions of 25 µM prepared in PBS were also incubated individually with the corresponding non-imprinted polymer, and the IF was calculated. Comparing the ΔI values obtained after the incubation of the kyn-MIP-SPCE and the corresponding non-imprinted polymer with l-Kyn and l-Trp solutions, the IF obtained for l-Kyn (10.5) was about seventeen times higher than the IF calculated for l-Trp (0.6). Indeed, as can be seen in Figure 7, the difference in terms of the ΔI, registered between the MIP and NIP sensors after incubation with l-Kyn, was very high, underlining the imprinting effect conferred from the imprinted polymeric layer electropolymerized on the sensor. On the other side, comparing the electrochemical response of MIP and NIP sensors incubated with l-Trp, no significant difference was observed (Figure 7).

### 3.4. Application of kyn-MIP-SPCE in Human Urine

The human urine composition obtains important information on human health thanks to the presence of important biomarkers that predict the presence of several pathologies. However, the high number of substances present in real samples can hinder instrumental analysis. For this reason, the pretreatment of human urine samples is a crucial step for the success of the analysis. Different methodologies were used to select the best and easiest procedure to purify human urine samples. In a first attempt, urine samples were centrifuged at 9000 rpm for 30 min. After that, 10 mL of supernatant was spiked with different concentrations of l-Kyn. After dilution (1/10, *v*/*v*), a drop (50 µL) of each solution was deposited on the kyn-MIP-SPCE working electrode and subjected to SWV analysis. Unfortunately, this procedure showed a linear correlation between the l-Kyn concentration and I_p_ only at high concentrations (250–1000 µM) of the analyte. For the aim of removing some compounds similar to the analyte that could hinder the electrochemical signal, such as Trp or other similar amino acids, the usage of a cartridge or a filter was considered. In a recent work, Fiore and co-workers used a solid phase extraction cartridge (Oasis HLB VAC RC) to purify serum samples before the electrochemical detection of l-Tyrosine, obtaining good results [35]. Starting from structural analogies between l-Tyrosine and l-Kyn, the same solid phase extraction cartridge was tested. In detail, 1 mL of human urine was loaded onto the cartridge, and the eluate was directly used for electrochemical measurements without dilution with the aim to enhance the I_P_ signal. However, in this case, no good results were obtained because the analyte was retained in the cartridge together with the interferences.

Finally, in this work a good and easy purification methodology was developed, filtering cold urine samples at 4 °C (stored for 2 h in the fridge) with a syringe filter of 0.2 µm. After spiking with l-Kyn at different concentrations (10–1000 µM), all solutions were diluted (1/10, *v*/*v*) with PBS 0.1 M (pH 7.4) and analyzed via SWV on the kyn-MIP-SPCE. The use of pretreated real samples required a higher time of incubation than standard solutions to obtain a stable signal. For this reason, after kinetic evaluation, 10 min were sufficient to obtain stable peak currents, and they were chosen for I_p_ values.

As can be seen in Figure 8, a linear correlation between the l-Kyn concentration and I_P_ was also obtained at concentrations lower than 250 µM (R^2^ = 0.9902), exceeding the limit that had been previously observed. The analytical method was validated by the determination of the LR, LOD, and LOQ. A good linearity was obtained in the range of 10–1000 µM with LOD and LOQ values of 1.5 and 5 µM, respectively. These results were compared with analytical performances of other modified electrochemical sensors and reported in Table 1.

## 4. Conclusions

In this work, a selective electrochemical sensor for l-Kyn detection was developed and successfully applied in real samples. Combining the advantage of molecularly imprinted technology and SPCE dispositions, an easy and low-cost electrochemical system was developed to selectively evaluate the concentration of l-Kyn in human urine samples. The best performance was obtained with the modified SPCE (kyn-MIP-SPCE) prepared with five scan cycles of *o*-PD electropolymerization. Morphological characterization of kyn-MIP-SPCE demonstrated the coverage of the SPCE working electrode with the polymeric layer since a slight attenuation of the corrugations typical of the bare SPCE was observed in SEM images. Additionally, [Fe(CN)_6_)]^3−/4−^ redox probe characterization has shown the absence of I_P_ associated with the redox probe on kyn-MIP-SPCE due to the growth of a non-conductive polymeric layer that hinders the electron transfer.

Kynetic study allowed us to find an optimum incubation period of 6 min for l-Kyn detection in standard solutions using the square wave voltammetry (SWV) method and giving a linear range between 10 and100 μM (R^2^ = 0.9924). Excellent selectivity and specificity results were found giving a very high selectivity factor α equal to 5.1 of kyn-MIP-SPCE calculated toward l-Trp, with an IF obtained for l-Kyn of about seventeen times higher than the IF calculated for l-Trp. Finally, the kyn-MIP-SPCE device allowed for l-Kyn detection in human urine samples, which was conducted only at a cold filtration, in a linear range of 10–1000 μM (R^2^ = 0.9902) with LOD and LOQ values of 1.5 and 5 µM, respectively. The developed disposable sensing system was demonstrated to be an interesting, fast, and cheap technology for the specific detection of the l-Kyn metabolite in biological fluids, which is of great relevance considering its recent correlation with SARS-CoV-2 disease progression.

## Figures and Tables

**Figure 1 polymers-16-00003-f001:**
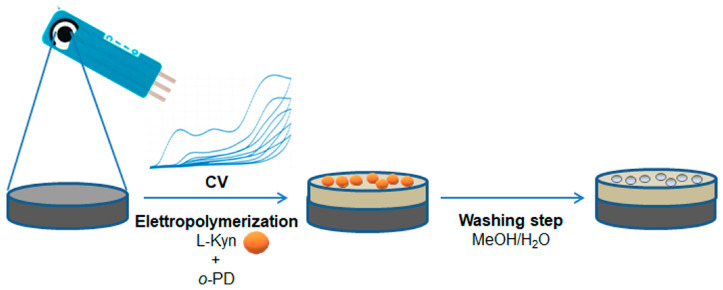
Schematic representation of kyn-MIP-SPCE preparation.

**Figure 2 polymers-16-00003-f002:**
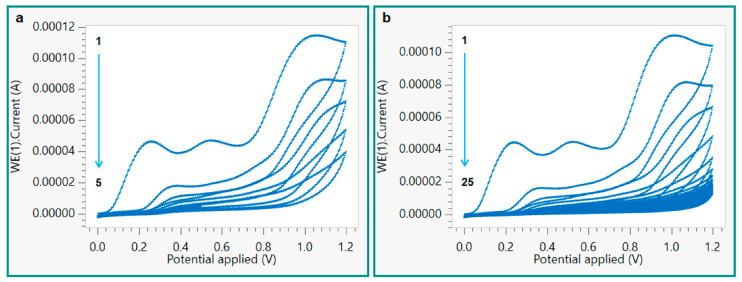
Electropolymerization of *o*-PD on a bare SPCE via CV conducted using 5 (**a**) and 25 (**b**) scan cycles.

**Figure 3 polymers-16-00003-f003:**
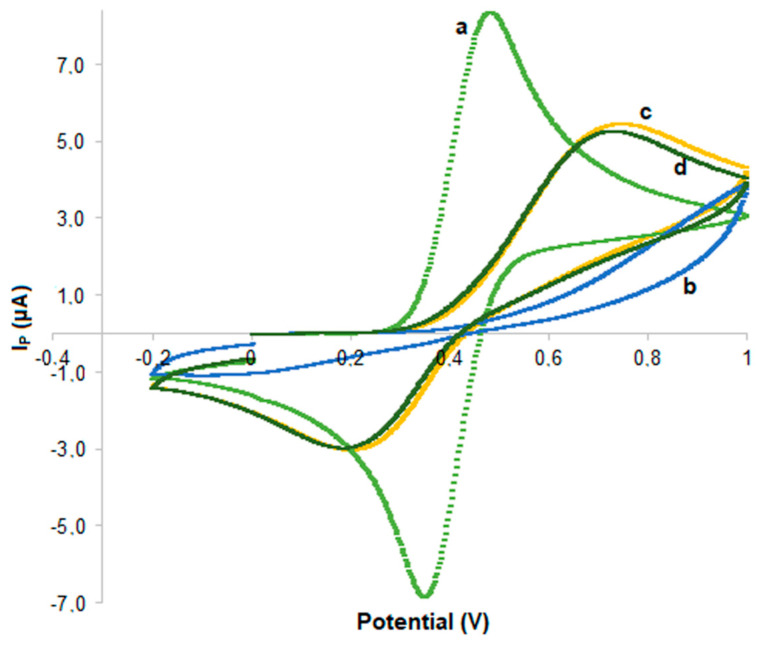
CV of each modification step of the SPCE assessed in 4 mM [Fe(CN)_6_)]^3−/4−^ redox probe with 100 mV/s: bare SPCE (**a**), kyn-MIP-SPCE before washing (**b**), kyn-MIP-SPCE after washing (**c**), kyn-MIP-SPCE after rebinding of l-Kyn (**d**).

**Figure 4 polymers-16-00003-f004:**
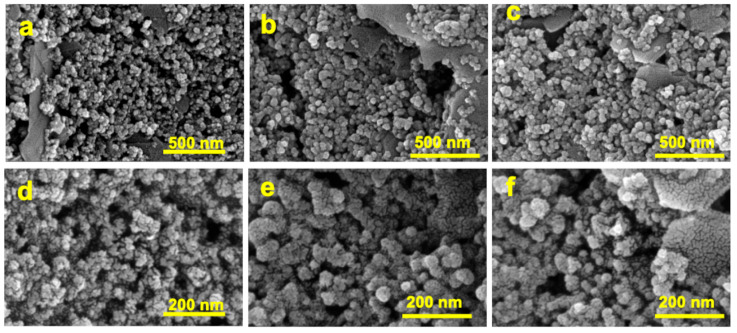
In-plane SEM images of work electrode of bare SPCE (**a**), kyn-MIP-SPCE (**b**), and NIP-SPCE (**c**) at 200,000× and bare SPCE (**d**), kyn-MIP-SPCE (**e**), and NIP-SPCE (**f**) at 500,000×.

**Figure 5 polymers-16-00003-f005:**
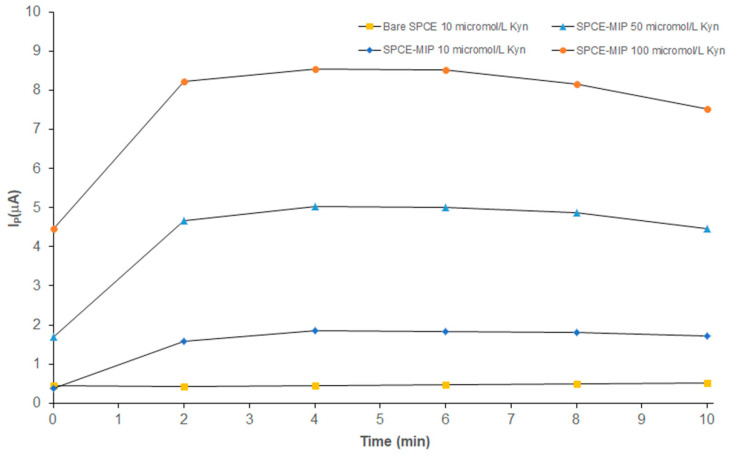
Kinetic analysis of bare SPCE incubated with 10 µM of l-Kyn standard solution (
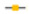
) and kyn-MIP-SPCE incubated with l-Kyn standard solutions at 10 µM (
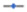
), 50 µM (
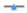
), and 100 µM (
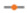
).

**Figure 6 polymers-16-00003-f006:**
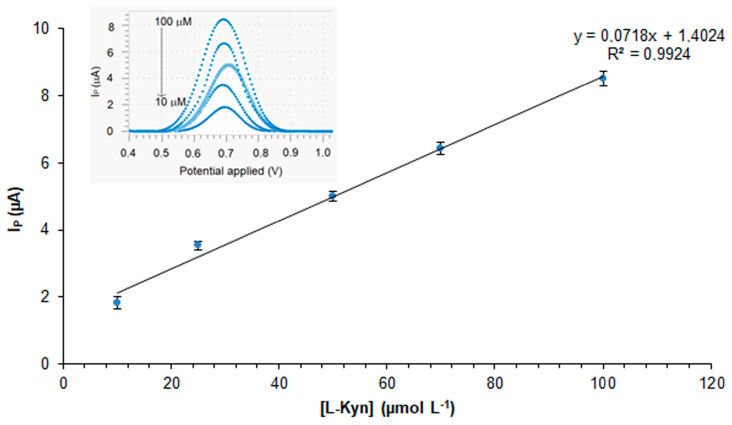
l-Kyn calibration curve (I_P_ vs. l-Kyn concentration) obtained via SWV analysis on kyn-MIP-SPCE.

**Figure 7 polymers-16-00003-f007:**
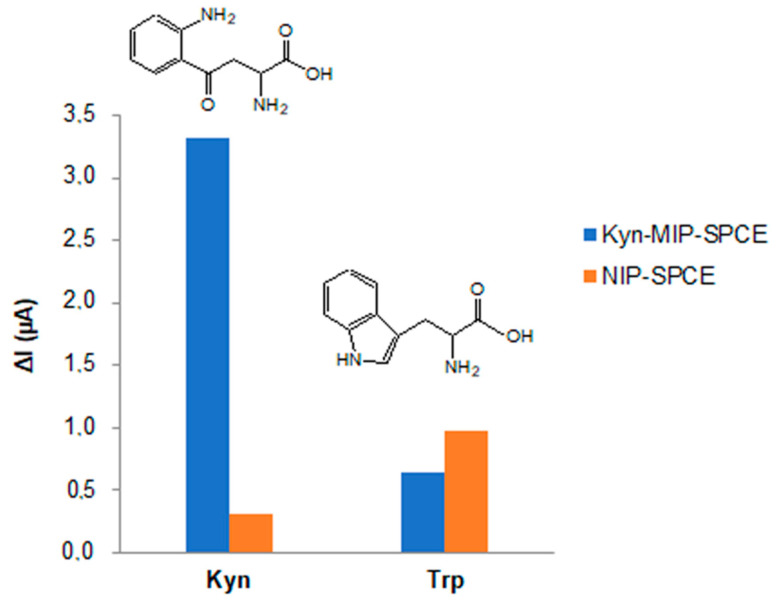
Selectivity studies performed incubating kyn-MIP-SPCE and NIP-SPCE individually with 25 µM of l-Kyn and l-Trp solutions.

**Figure 8 polymers-16-00003-f008:**
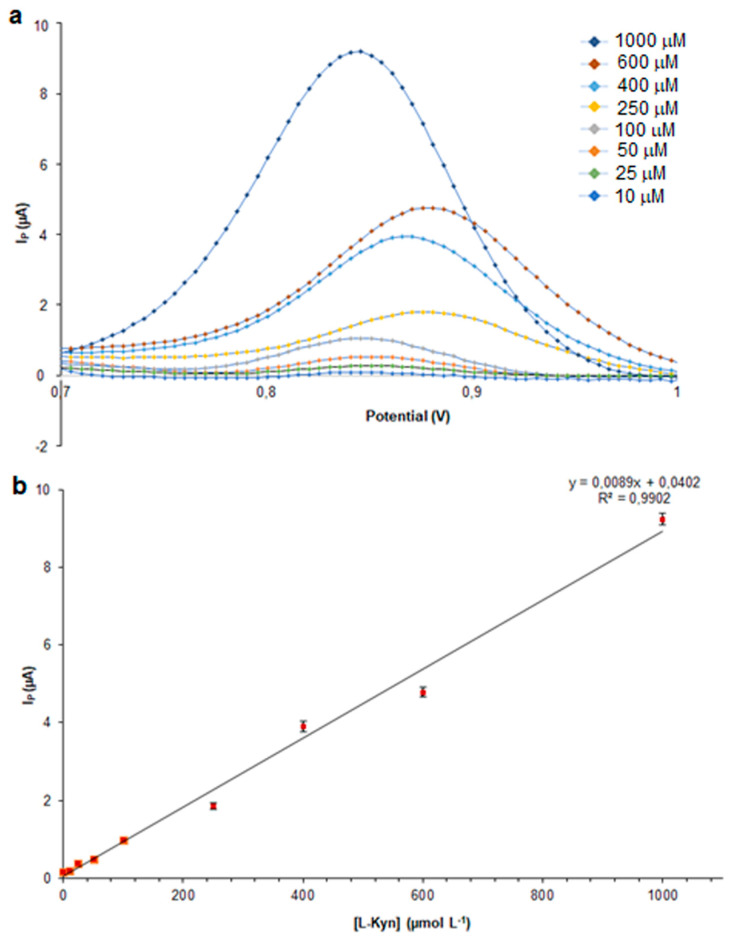
SWV of l-Kyn on kyn-MIP-SPCE (**a**) and plot of anodic I_P_ vs. l-Kyn concentration in human urine samples (**b**).

**Table 1 polymers-16-00003-t001:** Comparison of analytical performances of different modified SPEs.

SPE	Functional Monomer	Analyte	Matrix	LOD (µM)	LOQ (µM)	LR (µM)	Analytical Method	Incubation Time (min)	Ref.
MIP-SPCE	o-PD	Ecstasy	Urine	6.4	21	Up to 0.2	SWV	10	[27]
MIP-SPCE	o-PD	l-Kyn	Urine	1.5	5	10–1000	SWV	10	This work
GCE *	-	Amphetamine- like drugs	Serum	2.4	8.3	12–45	SWV	-	[36]
MIP-SPCE	Dopamine	Diclofenac	Water	0.07	0.2	0.1–10	DPV	30	[26]
SPE/rGO/AuNPs **	-	Trp	Buffer solution	0.39	1.32	0.5–500	DPV	-	[37]

* Glassy carbon electrode. ** SPE modified with graphene oxide/gold nanoparticles.

## Data Availability

Data are contained within the article.

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
