# Peer review of "Disposable Molecularly Imprinted Polymer-Modified Screen-Printed Electrodes for Rapid Electrochemical Detection of l-Kynurenine in Human Urine"

_polymers, 2023, doi:10.3390/polym16010003_

Round 1

Reviewer 1 Report

Comments and Suggestions for Authors

This manuscript is well written, after resolving the following issues, it can be considered published in Polymers.

(1)    What is “IF” in the abstract, its full name should be given.

(2)    Figure 5, it is better to use different shapes to show different test result

(3)    Figure 7, please give the chemical structures of the testing molecules.

(4)    A table to compare the detection performances (LOD, LR, detection time, real samples, etc.) of different methods particularly MIP-based methods is better provided.

Author Response

We wish to thank you for the evaluation and comments about our manuscript.

We have modified the manuscript taking into account your comments/suggestions.

All changes have been highlighted in the manuscript.

Please find below our answers to your comments:

Point 1.  What is “IF” in the abstract, its full name should be given.

Response 1. As suggested from the reviewer the full name of IF (Imprinting Factor) was explained in the abstract section.

Point 2. Figure 5, it is better to use different shapes to show different test result.

Response2. According to the reviewer's suggestion, the Figure 5 was modified using different shapes for each test as follow:

Point 3.  Figure 7, please give the chemical structures of the testing molecules.

Response 3. As suggested from the reviewer we added in Figure 7 the chemical structures of the testing molecules as follow:

      Point 4. A table to compare the detection performances (LOD, LR, detection time, real samples, etc.) of different methods particularly MIP-based methods is better provided.

Response 4. As suggested from the reviewer a table to compare detection performance of different SPE and MIP-SPE was added in the manuscript.

 Table 1. Comparison of analytical performances of different modified SPE

SPE

Functional monomer

Analyte

Matrix

LOD

(µM)

LOQ

(µM)

LR

(µM)

Analytical method

Incubation time (min)

Ref.

MIP-SPCE

o-PD

Ecstasy

Urine

6.4

21

Up to 0.2

SWV

10

27

MIP-SPCE

o-PD

l-Kyn

Urine

1.5

5

10-1000

SWV

10

This work

GCE*

-

Amphetamine-like drugs

Serum

2.4

8.3

12-45

SWV

-

36

MIP-SPCE

Dopamine

Diclofenac

Water

0.07

0.2

0.1-10

DPV

30

26

SPE/rGO/AuNPs **

-

Trp

Buffer solution

0.39

1.32

0.5-500

DPV

-

37

*Glassy Carbon Electrode

**Graphene oxide/gold nanoparticles modified SPE

Moreover, the following references were added in “References section”:

  1. Garrido, E.M.P.J.; Garrido, J.M.P.J.; Milhazes, N.; Borges, F.; Oliveira-Brett, A.M. Electrochemical oxidation of amphetamine-like drugs and application to electro- analysis of ecstasy in human serum. Bioelectrochemistry 2010 79, 77–83, https://doi.org/10.1016/j.bioelechem.2009.12.002.
  2. Nazarpour, S.; Hajian, R.; Sabzvari, M.H. A novel nanocomposite electrochemical sensor based on green synthesis of reduced graphene oxide/gold nanoparticles modified screen printed electrode for determination of tryptophan using response surface methodology approach. Microchem. J. 2020 154, 1046. https://doi.org/10.1016/j.microc.2020.104634

Reviewer 2 Report

Comments and Suggestions for Authors

Review: polymers-2756838.

Title: Disposable molecularly imprinted polymers-modified screen-printed electrodes for rapid electrochemical detection of L-kynurenine in human urine.

In this manuscript Authors present investigations of molecularly imprinted polymers (MIPs) modified screen-printed electrodes for the determination of L-kynurenine, an endogenous metabolite of L-tryptophan and a potential biomarker of several immunological disorders. The MIP component was composed from L-kynurenine as a template and o-phenylenediamine that was electropolymerized on a bare screen-printed carbon electrode. The capability to determine the L-kynurenine was verified in the analysis of L-kynurenine in human urine samples. The process required the optimization of analytical conditions. In my opinion, the manuscript could be interesting for the Readers of the Journal since selective electrochemical methods of measurements are requested nowadays by the analytical chemistry. However, a few drawbacks should be addressed by Authors at this stage of evaluation:

1/ The novelty shall be convincingly justified from a broader perspective. The statement that ‘…for the first time a combination of SPE electrochemical sensors and MIP advantages to the aim to obtain a disposable and portable device for specific L-Kyn recognition…’ could be considered as insufficient. The electrochemical sensors based on MIPs are described in a plethora of papers. Authors shall find also another factors e.g. the evaluation of new analytical method to enhance the novelty.

2/ Another paper of the group that was published recently (see: J. Sep. Sci. 2018, 41, 3204). Please, provide a comparable discussion of results. It could be interesting to compare the imprinting efficacy of poly(methacrylic acid-co-trimethylpropane trimethacrylate) imprinted by L-kynurenine to o-phenylenediamine polymer imprinted by L-kynurenine.

3/ The critical discussion of the selectivity studies in the context of above mentioned paper  shall be completed. Please, compare and discuss the selectivity of both materials towards L-kynurenine  and L-tryptophan (see: J. Sep. Sci. 2018, 41, 3204, Figure 3 and Figure 7 in this manuscript). It was stated in Conclusions that ‘…very high selectivity factor α equal to 5.1 of kyn-MIP-SPCE calculated toward L-Trp with also an IF obtained for L-Kyn of about seventeen times higher than the IF calculated for L-Trp…’. Does it means that the o-phenylenediamine polymer is more selective than poly(methacrylic acid-co-trimethylpropane trimethacrylate)?

4/ It was revealed that the complex sample of human urine required to be pre-treated prior to the electrochemical analysis. Please, explain the use of commercial solid phase sorbent of Oasis HLB VAC RC instead of previously developed SPE imprinted by L-kynurenine?

5/ The utility of the sensor could be affected by the repeatability and reproducibility. Please, comment it and if necessary, extend the experiments.

I hope that above mentioned suggestions will strengthen the scientific value of the manuscript.

Therefore, in my opinion, major revision is required before final decision of the Editor.

Author Response

Review: polymers-2756838.

We wish to thank you for the evaluation and comments about our manuscript.

We have modified the manuscript taking into account your comments/suggestions.

All changes have been highlighted in the manuscript.

Please find below our answers to your comments:

Point 1.  The novelty shall be convincingly justified from a broader perspective. The statement that ‘…for the first time a combination of SPE electrochemical sensors and MIP advantages to the aim to obtain a disposable and portable device for specific L-Kyn recognition…’ could be considered as insufficient. The electrochemical sensors based on MIPs are described in a plethora of papers. Authors shall find also another factors e.g. the evaluation of new analytical method to enhance the novelty.

Response 1. As suggested from the reviewer more factors have been added at the end of section 1 (Introduction), and the following sentence have been added as follow:

paragraph 1: “In the present work, for the first time a combination of SPE electrochemical sensors and MIP advantages to the aim to obtain a disposable and portable device for specific l-Kyn recognition, has been reported. Some key aspects have been optimized such as the choice of a suitable and effective imprinted polymer and also a reliable new analytical method based on Square Wave Voltammetry (SWV) for urine samples analysis of l-Kyn as well an easy urine pre-treatment.  The SPE was used as a sensor for specific l-Kyn recognition after its modification through the electrochemical in situ MIP synthesis. o-PD monomer electropolymerization on the working electrode surface of a commercial SPCE in the presence of l-Kyn molecule as template was carried out. The optimized modified sensor specific for l-Kyn detection (Kyn-MIP-SPCE) has been characterized by cyclic voltammetry (CV), using [Fe(CN)6)]3-/4- as a redox probe, and scanning electron microscopy (SEM). After the optimization of various experimental parameters, such as the number of CV electropolymerization cycles, electrochemical measurements method, incubation period, l-Kyn has been detected firstly in standard solution by Square Wave Voltammetry (SWV).

Point 2.  Another paper of the group that was published recently (see: J. Sep. Sci. 2018, 41, 3204). Please, provide a comparable discussion of results. It could be interesting to compare the imprinting efficacy of poly(methacrylic acid-co-trimethylpropane trimethacrylate) imprinted by L-kynurenine to o-phenylenediamine polymer imprinted by L-kynurenine.

Response 2. We added a comparable discussion of the imprinted polymer described in our previous paper (2018) and the Kyn-MIP-SPCE described here both for l-Kyn recognition only at the beginning of Results and Discussion section where we explained why we chose  o-phenylenediamine instead to continue with  poly(methacrylic acid-co-trimethylpropane trimethacrylate):

paragraph 3.1: “In a previous work we prepared an imprinted polymer for l-Kyn recognition by using  methacrylic acid and trimethylpropane trimethacrylate as functional monomer and crosslinker respectively and excellent recognition properties were found when the MIP was used as sorbent for solid phase extraction of urine samples [22]. In the present work, a first attempt to use the same MIP to modify the working electrode of a SPCE was made. However, to overcome some drawbacks typical of MIP sensors obtained from the deposition of a conventional polymer such as solvent consuming, the difficulty to control thickness, morphology and the reproducibility as well to have an effective electron transfer interface, we addressed our research towards the direct MIP synthesis on the work electrode surface  through the electropolymerization technique. o-PD has been the most used functional monomer for MIP-based sensor fabrication by electropolymerization process thanks to its high stability [27, 30, 33].”

In order to avoid confusion, we preferred not to add a comparable discussion of results into the text. In fact, we guess that the MIP systems can not be directly compared since they are used for different applications. To support this idea we can consider the difficulty of using the MIP prepared in our previous work in electrochemical sensor application (see the above discussion).

Point 3. The critical discussion of the selectivity studies in the context of above mentioned paper  shall be completed. Please, compare and discuss the selectivity of both materials towards L-kynurenine  and L-tryptophan (see: J. Sep. Sci. 2018, 41, 3204, Figure 3 and Figure 7 in this manuscript). It was stated in Conclusions that ‘…very high selectivity factor α equal to 5.1 of kyn-MIP-SPCE calculated toward L-Trp with also an IF obtained for L-Kyn of about seventeen times higher than the IF calculated for L-Trp…’. Does it means that the o-phenylenediamine polymer is more selective than poly(methacrylic acid-co-trimethylpropane trimethacrylate)?

Response 3 Even if the reviewer suggested to complete the discussion about selectivity studies comparing our  latest results with our previous results on a different MIP system for the same template (L-Kyn) and competitive molecule (L-Trp), as observed in the previous point, we believe that is not correct a direct comparison to define which one is the best system but we added  the following sentence in paragraph 3.3 about selectivity studies:

paragraph 3.3: “The low ΔI recorded after l-Trp incubation with kyn-MIP-SPCE was due to the formation of non-specific hydrogen bonds between l-Trp and the surface of the polymeric layer. Indeed, the selectivity factor α equal to 5.1 of kyn-MIP-SPCE calculated toward l-Trp was very high demonstrating the high selectivity of the modified sensor for l-Kyn. Comparing this last result with another MIP for l-kyn reported in literature [22], a similar selectivity factor was obtained (α = 4)  demonstrating that each developed system is suitable for l-Kyn recognition in its specific application.

Point 4.  It was revealed that the complex sample of human urine required to be pre-treated prior to the electrochemical analysis. Please, explain the use of commercial solid phase sorbent of Oasis HLB VAC RC instead of previously developed SPE imprinted by L-kynurenine?

Response 4. As suggested from the reviewer we clarified our choice adding the following sentence into the text but we do not refer to our previously developed SPE imprinted by L-kynurenine since our aim was to use an easy procedure (one single step) enables to retain the interferences while SPE imprinted by L-kynurenine works in two steps, firstly L-Kyn is retained into the cartridge and then it needs to be removed:

paragraph 3.4: “To the aim to remove some compounds  similar to the analyte such as L-Trp or other similar amino acids  that could hinder the electrochemical signal, the usage of a cartridge or a filter was thought. In a recent work, Fiore and co-workers used a solid phase extraction cartridge (Oasis HLB VAC RC) to purify serum samples before electrochemical detection of l-Tyrosine obtaining good results [35]. Starting from structural analogies between l-Tyrosine and l-Kyn, the same solid phase extraction cartridge was tested. In detail, 1 ml of human urine was loaded on the cartridge and the eluate was directly used for electrochemical measurements without dilution with the aim to enhance IP signal. However, also in this case, no good results were obtained because the analyte was retained into the cartridge together with the interferences.

Point 5.  The utility of the sensor could be affected by the repeatability and reproducibility. Please, comment it and if necessary, extend the experiments.

Response 5. All rebinding experiments were repeated three times but we forgot to underline it into the text  thus into section 2 (Experimental) we added the sentence “All experiments were repeated three times” into the proper paragraphs. Moreover it is worth noting that each rebinding experiment (and electrochemical measurement) was made with a new kyn-MIP-SPCE or NIP-SPCE  prepared for each experiment by using the same electropolymerization procedure and this confirm the repeatability and reproducibility of the developed system that have to work as a disposable system.

Round 2

Reviewer 2 Report

Comments and Suggestions for Authors

Review: polymers-2756838-R1.

Title: Disposable molecularly imprinted polymers-modified screen-printed electrodes for rapid electrochemical detection of L-kynurenine in human urine.

In this revised manuscript, Authors have made corrections and explanations according to Referee comments. In my opinion, the manuscript in current form could be considered for acceptance.